# Depolymerisation of Kraft Lignin by Tailor-Made Alkaliphilic Fungal Laccases

**DOI:** 10.3390/polym15224433

**Published:** 2023-11-16

**Authors:** David Rodríguez-Escribano, Felipe de Salas, Rocío Pliego, Gisela Marques, Thomas Levée, Anu Suonpää, Ana Gutiérrez, Ángel T. Martínez, Petri Ihalainen, Jorge Rencoret, Susana Camarero

**Affiliations:** 1Centro de Investigaciones Biológicas Margarita Salas, Consejo Superior de Investigaciones Científicas (CSIC), 28040 Madrid, Spain; davidre04@gmail.com (D.R.-E.); atmartinez@cib.csic.es (Á.T.M.); 2Instituto de Recursos Naturales y Agrobiología de Sevilla, Consejo Superior de Investigaciones Científicas (CSIC), 41012 Sevilla, Spain; gisela@irnase.csic.es (G.M.); anagu@irnase.csic.es (A.G.); 3MetGen Oy, 20780 Kaarina, Finlandanu@metgen.com (A.S.); petri.ihalainen@metgen.com (P.I.)

**Keywords:** kraft lignin, alkaliphilic laccase, chemical analyses, depolymerisation, membrane separation system

## Abstract

Lignins released in the black liquors of kraft pulp mills are an underutilised source of aromatics. Due to their phenol oxidase activity, laccases from ligninolytic fungi are suitable biocatalysts to depolymerise kraft lignins, which are characterised by their elevated phenolic content. However, the alkaline conditions necessary to solubilise kraft lignins make it difficult to use fungal laccases whose activity is inherently acidic. We recently developed through enzyme-directed evolution high-redox potential laccases active and stable at pH 10. Here, the ability of these tailor-made alkaliphilic fungal laccases to oxidise, demethylate, and depolymerise eucalyptus kraft lignin at pH 10 is evidenced by the increment in the content of phenolic hydroxyl and carbonyl groups, the methanol released, and the appearance of lower molecular weight moieties after laccase treatment. Nonetheless, in a second assay carried out with higher enzyme and lignin concentrations, these changes were accompanied by a strong increase in the molecular weight and content of β–O–4 and β–5 linkages of the main lignin fraction, indicating that repolymerisation of the oxidised products prevails in one-pot reactions. To prevent it, we finally conducted the enzymatic reaction in a bench-scale reactor coupled to a membrane separation system and were able to prove the depolymerisation of kraft lignin by high-redox alkaliphilic laccase.

## 1. Introduction

Lignin is the most abundant aromatic biopolymer on earth and the main renewable source of aromatic molecules (i.e., phenols) as sustainable alternatives to petroleum-derived chemicals [1]. Lignin is primarily composed of three phenylpropanoid units—syringyl (S), guaiacyl (G), and p-hydroxyphenyl (H)—linked through ether and carbon–carbon bonds such as β–O–4, β–β, etc. [2,3]. The lignin matrix covers cellulose microfibrils in the secondary wall of plant cells, providing protection against microbial attack and structural integrity [4].

Due to its high recalcitrance, lignin must be removed under harsh extraction conditions for the industrial conversion of cellulose to paper pulp or liquid fuels. The leftover lignins are solubilised in the black liquors and generally burnt to generate energy for the mill [5]. However, lignin’s hydrophobic, thermal and binding properties, and chemical aromatic composition are ideal properties to be converted into bio-based materials for advanced applications and chemicals [5,6]. Today, the pulp and paper industries generate about 60 million tons of lignin per year and an additional 225 million tons of lignin per year are estimated to be obtained by 2030 [7]. Since kraft pulping represents over 85% of the global pulping capacity, kraft lignins constitute the largest lignin stream by volume. Despite this and the recent advances made in their supply through the LignoBoost^®^ (Valmet, Espoo, Finland) [8] or LignoForce™ (FPInnovations, Quebec, QC, Canada) [9] processes, kraft lignins are still largely commercially unexploited.

Kraft lignins have a strong phenolic nature as a result of the cleavage of most ether bonds during pulping [10,11]. They can therefore be used as green substitutes of current phenolic binder systems, which may be based on lignin activation by enzymatic processes [12,13]. However, their deconstruction to obtain low-molecular-weight compounds remains a challenge. The condensation of lignin fragments during the pulping process produces highly recalcitrant C–C linkages that make difficult the depolymerisation of kraft lignins [6].

Laccases (EC 1.10.3.2; p-diphenol oxygen oxidoreductases) are multi-copper-oxidases that use molecular oxygen to catalyse the oxidation of a great variety of aromatic substrates including lignin [14]. Their phenol oxidase activity makes laccases particularly suited to modifying kraft lignins. White-rot basidiomycetes are the main agents responsible for lignin decay in nature and valued sources of high-redox potential laccases. The superior oxidising capabilities of these fungal laccases, compared to those of low-redox potential laccases from plants or bacteria, are of special biotechnological relevance for the deconstruction of kraft lignins to produce aromatic compounds. However, the alkaline conditions in which kraft lignins are soluble make their modification by fungal laccases, which have naturally evolved to oxidise lignin in acidic conditions and mild temperatures, difficult. To overcome this limitation, we recently engineered a set of high-redox potential fungal laccases with alkaliphilic properties [15]. These novel enzymes are tested in the present study as biocatalysts to depolymerise kraft lignin.

Another potential application of these tailor-made enzymes is in the bleaching of kraft pulps. For instance, their integration in the ECF (elemental chlorine free) bleaching sequence of hardwood kraft pulps (at 65 °C, pH 8.5) allowed us to improve the degree of pulp brightness, making it possible to significantly reduce the loading of ClO_2_ as a bleaching agent [15]. These laccases can also be applied in the manufacturing of medium-density fibreboards, added directly to the wood chips before refining, to reduce the energy required to obtain wood fibres, as well as in the formulation of lignin-based phenolic resins used as adhesives for the panel boards. The addition of enzymes to the formulation improves the resin-formation process and reduces the amount of formaldehyde needed [15]. The applicability of alkaliphilic laccases goes beyond the wood-conversion sector. It ranges from the textile industry (e.g., in fabric dyeing or finishing, or to remove colour from textile waste waters) to hair dyeing and chemical synthesis [16,17,18,19].

Upon oxidation of lignin units by laccase, demethylation, Cα or Cβ oxidation, and cleavage of Cα–Cβ side-chains and β–O–4 aryl bonds take place, but simultaneous radical coupling makes it difficult to monitor the progress of the reaction [20]. The repolymerisation event that counteracts the breakdown of lignin by laccases is markedly more pronounced at lower pH [21]. Taking all this into account and the higher solubility of lignin in alkaline media, a lignin-refining technology (METNIN™) was developed based on the use of a bacterial alkaliphilic laccase combined with a membrane cascade system with different cut-offs to prevent lignin repolymerisation. The commercial process allows us to obtain a set of lignin fractions with consecutively lower molecular weights and reduced polydispersity [22].

In this work, we evaluate the ability of the aforementioned alkaliphilic high-redox potential fungal laccases [15] to oxidise and depolymerise eucalyptus kraft lignin. The co-occurrence of depolymerisation and repolymerisation in one-pot reactions leads to changes in the lignin structure that we analysed with different techniques such as 2D-NMR, ^31^P-NMR, GPC, and GC/MS–. Finally, we demonstrate the depolymerisation of lignin by a high-redox potential laccase using a bench-scale membrane separation system based on the METNIN™ principle.

## 2. Materials and Methods

### 2.1. Material

The eucalyptus kraft lignin used in this study was provided by the Centre Technique du Papier (Grenoble, France). This lignin was isolated from the black liquors of The Navigator Company kraft pulp mill.

### 2.2. Flask Production and Purification of Laccases

The *S. cerevisiae* clones expressing laccase were grown in duplicate in 100 mL or 1 L flasks with 30 mL or 300 mL of expression medium, respectively, containing 1 M CuSO_4_ and 3% ethanol [23]. The laccase activity secreted in the liquid cultures was monitored over time by measuring the oxidation of 3 mM 2,2′-azino-bis(3-ethylbenzothiazoline-6-sulfonic acid) (ABTS) in 50 mM citrate phosphate buffer pH 3, 3 mM 2,6-dimethoxyphenol (DMP), 100 mM sodium phosphate buffer (pH 6 and 8), and 9 mM guaiacol in 100 mM Britton–Robinson buffer pH 9 [15] with UV-1900 Shimadzu spectrophotometer. After 3 days of fermentation, the cultures were centrifuged, filtered, and concentrated as previously described. Enzyme purification was carried out using high pressure liquid chromatography (HPLC) in three chromatographic steps: two anion-exchange steps using a HiPrep-QFF 16/10 column ( Cytiva, Uppsala, Sweden) in a 100 mL gradient of 0–40% elution buffer and a MonoQ-HR 5/50 column (Cytiva, Uppsala, Sweden)in a 30 mL gradient of 0–25% elution buffer, followed by size exclusion chromatography with a Superdex 75 (Cytiva, Uppsala, Sweden). The fractions containing laccase activity (with 3 mM ABTS in 50 mM citrate-phosphate pH 3) were pooled, dialysed in Tris-HCl pH 7, and concentrated after each chromatographic step. The isolation of the enzyme was confirmed using the electrophoretic mobility in SDS-PAGE (12% acrylamide) stained with Coomassie brilliant blue.

### 2.3. Kraft Lignin Oxidation

#### 2.3.1. Effect of pH and Enzyme Variant on Treatment of Lignin at Low Concentration

Three alkaliphilic fungal laccase variants, Li10, Li11, and C-LeB, recently developed using directed evolution [15], were compared for the oxidation of eucalyptus kraft lignin. In order to evaluate the effect of pH on the enzymatic transformation of lignin, eucalyptus kraft lignin (0.5 g/L) was solubilised in 20 mM B&R buffer pH 9 or pH 10 and treated with the three purified laccase variants using 0.2 U/g of lignin (50 mL final reaction volume) for 2 and 24 h at 30 °C and 180 rpm. Units are referred to as the amount of enzyme that catalyses the oxidation of 1 µmol of ABTS to its cation radical (ε_418_, 36,000 M^−1^ cm^−1^) per min. The phenolic and carbonyl contents of the lignin samples were determined spectrophotometrically in triplicate using the Folin–Ciocalteu reaction (FCR Abs. 760 nm) and Brady reagent (2,4-Dinitrophenol, Abs. 450 nm) purchased from Sigma-Aldrich (Madrid, Spain), respectively [24], and the molecular weight (Mw) distribution profiles using size-exclusion chromatography (SEC). For SEC analyses, the lignin samples were completely solubilised in 100 mM NaOH (Merck, Darmstadt, Germany) pH 11.6, centrifuged (13,400 rpm), and injected in a Superdex 75 column pre-equilibrated with 20 mM Britton–Robinson buffer (pH 11.6). The absorbances at 260 and 280 nm were monitored throughout the chromatographic run.

#### 2.3.2. Demethylation Assay

The methanol released after lignin oxidation by laccase was measured using its transformation to formaldehyde by an alcohol oxidase from *Komagataella phaffii* (*Pichia pastoris*) and the photometric measurement of the reaction with the Purpald reagent (4-amino-3-hydrazino-5-mercapto-1,2,4-triazole) purchased from Sigma-Aldrich, (Madrid, Spain). The demethylation assay [22] was adapted to 96-well plates. A stock solution of 34 mM Purpald reagent in 0.5 mM NaOH and a methanol standard curve with a concentration ranging from 0 to 0.5 mM were prepared. In each well, 28 µL of diluted lignin sample or methanol standard was added to 28 µL of 1 U/mL alcohol oxidase in 100 mM sodium phosphate pH 7. Negative samples were prepared by adding buffer without alcohol oxidase. The reactions were carried out in triplicate. The samples were incubated for 10 min at 30 °C under 250 rpm agitation. Next, 56 µL of the Purpald stock solution was added to the reaction and the samples were incubated for 30 min at 30 °C, 250 rpm. The reactions were stopped by adding 168 µL H_2_O and incubated for 1 min. The samples were measured at 560 nm using a SpectraMax M2 microplate reader (Molecular Devices, Sunnyvale, CA, USA). The methanol concentration of the samples was calculated using the values obtained from the methanol standards.

#### 2.3.3. Enzymatic Treatment of Lignin at High Concentration

Eucalyptus kraft lignin (25 g/L) was treated with 80 U of purified alkaliphilic (C-LeB) laccase per g of lignin in 20 mM B&R buffer pH 10 (50 mL final reaction volume) at 30 °C. The same conditions without the enzyme served as the control lignin. Samples (100 µL) were taken at 2 and 24 h to determine their phenolic and carbonyl contents and Mw distribution profiles as mentioned above. After 24 h of reaction the aqueous (A) and solid (S) phases of laccase-treated (L) and control (C) lignins were dried, resuspended in 50 mL of 37% HCl, and centrifuged for 20 min at 6000× *g* at 4 °C. The precipitated fractions were frozen and lyophilised. The weight of lignin in the acid-precipitated samples was measured with gravimetric methods whereas the content of acid-soluble lignin was spectrophotometrically determined using the following function:LS(%) = V × E_205_/1100 M(1)

LS(%): soluble lignin as a percentage of total lignin; V: total volume; E_205_: absorbance of lignin at 205 nm; M: total lignin mass.

The effect of shorter reaction times and different enzyme: lignin ratios on the phenolic content (measured with FCR) were analysed in small reactions (2 mL final volume) using 6, 30, 60, and 180 U of C-LeB laccase per g of lignin at 0.5 g/L lignin concentration of the different samples.

### 2.4. Gel Permeation Chromatography (GPC) Analysis

For the GPC analyses, kraft lignin samples were previously acetylated (pyridine/acetic anhydride) and dissolved in tetrahydrofuran (THF). GPC was performed on a Shimadzu Prominence-i LC-2030 3D GPC system (Shimadzu, Kyoto, Japan) equipped with a photodiode array (PDA) detector, using the following conditions: column, PLgel 5 µm MIXED-D, 7.5 × 300 mm (Agilent Technologies, Cheadle, UK); THF as eluent; flow rate, 0.5 mL min^−^^1^; temperature, 40 °C; sample detection, PDA response at 280 nm. The data acquisition and computation used LabSolution GPC software version 5.82 (Shimadzu, Kyoto, Japan). The molecular weight calibration was via polystyrene standards (Mw range from 5.8 × 102 up to 3.24 × 106 Da, Agilent Technologies, Cheadle, UK).

### 2.5. Nuclear Magnetic Resonance Analyses

Lyophilised samples from the acid-precipitated A and S fraction of the C and L lignins were analysed with nuclear magnetic resonance (NMR), including ^31^P-NMR and two-dimension heteronuclear single quantum coherence (2D-HSQC) NMR experiments.

For the 2D-HSQC measurements, kraft lignin samples (~30 mg) were transferred into a 5 mm NMR tube, dissolved in 0.5 mL of deuterated dimethyl sulfoxide (DMSO-*d*_6_), and analysed on a Bruker Avance III 500 MHz (Bruker, Karlsruhe, Germany) spectrometer fitted with a 5 mm TCI (triple cryoprobe inverse) probe. The 2D-HSQC NMR spectra were acquired at 300 K using an adiabatic pulse sequence (hsqcetgpsisp.2). The spectra were acquired from 10 to 0 ppm in ^1^H using 1676 data points for an acquisition time (AQ) of 145 ms, an interscan delay (D1) of 1 s, and from 165 to 0 ppm in ^13^C using 256 increments of 32 scan, for a total experimental time of 2 h 40 min. The ^1^*J*_CH_ coupling constant used was 145 Hz. The residual DMSO central peak (δ_C_/δ_H_ 39.5/2.49) was used as an internal reference. HSQC correlation peaks were assigned according to the literature [25], and the quantitation of lignin units and linkages was performed as described elsewhere [26].

For the ^31^P-NMR analysis, the lignin samples were phosphitylated as previously described [25]. Briefly, 30 mg of dry lignin was transferred into a 5 mm-id NMR tube and was dissolved in pyridine/CDCl_3_ (300 µL, 1.6/1.0, *v*/*v*). Stock solutions of the internal standard (cholesterol, 20.9 mg/mL or N-Hydroxy-5-norbornene-2,3-dicarboximide, 9.7 mg/mL) and relaxation reagent (chromium (III) acetylacetonate, 10.5 mg/mL) were prepared separately using the same pyridine/CDCl_3_ solvent mixture, and 150 μL and 75 μL, respectively, were added to the NMR tube containing the lignin mixture. Then, 75 μL of derivatisation reagent (2-chloro-4,4,5,5-tetramethyl-1,3,2- dioxaphospholane) was also added. The ^31^P-NMR spectra were acquired on a Bruker Avance Neo 500 MHz NMR spectrometer using the standard phosphorus pulse program of Bruker “zgig”, with a relaxation delay of 5 s and 256 accumulated scans. The chemical shift of the signal arising from the product from residual water and 2-chloro-4,4,5,5-tetramethyl-1,3,2-dioxaphospholane (at 132.2 ppm) was used as reference. Each lignin sample was analysed in triplicate.

### 2.6. GC/MS

The acid-soluble lignin samples (Aas and Sas) were extracted twice with 2 volumes of chloroform. The extracted solution was frozen in order to separate the chloroform phase, air-dried, and concentrated using rotary evaporation. Duplicate aliquots of this solution were withdrawn and 100 µg/mL of ethylvanillin in the same solvent was added as an internal standard. Before GC–MS analysis, samples were trimethylsilylated (TMS) at 80 °C for 20 min using 0.2 mL of the BSTFA reagent. After derivatisation, the low molecular weight compounds extracted from the control sample and the laccase treatments were identified and quantified using a gas chromatograph equipped with an HP-5MS column (30 m × 0.25 mm internal diameter; 0.25 μM film thickness) coupled to a quadrupole mass detector (GC–MS system 7980A-5975C, Agilent Technologies).

### 2.7. Membrane Separation System at Bench Scale

A scheme illustrating the membrane separation system setup to carry out the enzymatic reaction at bench scale is provided in Appendix A. Prior to the trials, lignin was solubilised at ~50 g/L in 0.1 M NaOH (108.1 g lignin in 2 L water), mixed for 30 min at room temperature, and centrifuged at 6000× *g* for 20 min. The collected supernatant was subsequently dried in an oven at 105 °C and stored at room temperature until further use. Then, the dried lignin was dissolved in water to a final concentration of 25 g/L. The reactions were run in 7.5 L bioreactors (Labfors 5, Infors HT, Bottmingen, Switzerland) equilibrated to 30 °C and pH adjusted to 10 for the fungal laccase (Li10), and 40 °C and pH 10.6 for the bacterial laccase. Then, the enzyme was added to the reactor: 5 U/g of dry lignin for the fungal laccase and 25 or 100 U/g of dry lignin for the bacterial laccase. The control reaction was performed under same conditions without the enzyme. The reaction was continued for 2 h with the fungal laccase and 16 h with the bacterial laccase, both with constant aeration of 1 L/min (with a sparger) under stirring (400 rpm). For separation of lignin fractions (retentate and permeate), ultrafiltration (UF) was performed using two tangential flow polyethersulfone membrane cassette units of 50 kDa cutoff (Vivaflow 200, Sartorius, Göttingen, Germany) with 3 × 300 mL diafiltration rounds in the end. The UF units were used according to the manufacturer’s protocol (1.5 bar pressure, 200–400 mL/min retentate flow/module). At the end of the process, the retentate and permeate samples were collected and dried in an oven at 60 °C before chemical analyses.

## 3. Results and Discussion

### 3.1. Effect of pH on Lignin Transformation by Three Alkaliphilic Fungal Laccase Variants

Three novel tailor-made alkaliphilic fungal laccase variants recently developed in our lab, Li10, Li11, and C-LeB [15], were compared here for the oxidation of eucalyptus kraft lignin (0.5 g/L). The enzymatic reactions were carried out at pH 9 (Figure 1A,C,E) and 10 (Figure 1B,D,F).

After a 24 h reaction, changes in the content of phenolic OH groups and carbonyl groups and in the lignin Mw distribution profile were determined (Figure 1). The lignin phenolic content strongly decreased (Figure 1A) and the carbonylic content increased (Figure 1C) after the enzymatic treatment at pH 9, with all laccases producing similar results. In contrast, after laccase treatment at pH 10 the phenolic content slightly increased (Figure 1B) together with the content of carbonyl groups (Figure 1D); the effect of the three variants was again comparable. Laccase strongly modified the lignin Mw profile at pH 9, with an important displacement towards higher Mw (Figure 1E). At pH 10, a smaller shift towards higher Mw was observed, but, interestingly, new peaks of lignin molecules with lower Mw emerged (Figure 1F), which were not observed at pH 9.

Kraft lignins are characterised by a high content of free phenolic hydroxyl groups due to the cleavage of most aryl–ether bonds (mainly β–O–4) by the pulping conditions (Na_2_S/NaOH and up to 170 °C) [11]. The significant decrease in the phenolic OH content of the eucalyptus kraft lignin after laccase treatment at pH 9 (or at pH < 9 observed in previous assays; data not shown), along with the strong increase in the lignin Mw, can be associated with the easy oxidation of phenolic OH groups by laccase and the strong tendency of the resulting phenoxyl radicals to couple, leading to the formation of new linkages in condensation reactions [27]. Quite the opposite, the small increase in phenolic content after lignin oxidation by laccase at pH 10 can be the result of bond cleavage [28], in agreement with the reported impact of pH on the balance between cleavage and polymerisation reactions [21]. The appearance of new lignin moieties with reduced Mw at pH 10 also evidenced depolymerisation, which is favoured at more alkaline pH [22]. On the other hand, the increase in carbonyl groups in all laccase-treated lignins was the result of the oxidation of the side chains of the lignin units and the subsequent formation of quinones [21].

Finally, as demethylation of lignin substituents can also generate new phenolic OH groups, we investigated this event using the Purpald assay [22]. The methanol released from the laccase-treated lignin increased with the reaction time compared to the control, indicating that lignin was demethylated by the enzyme (Figure 2).

### 3.2. Lignin Transformation at High Concentration with Alkaliphilic Fungal Laccase

Next, we carried out another lab assay in which lignin concentration in the reaction was raised to 25 g/L (to better match industrial conditions) and to provide sufficient lignin for subsequent chemical analyses. The concentration of laccase was also significantly increased. We used the C-LeB variant, but similar results are expected for the other two fungal enzymes owing to the similar results obtained in the previous assay (Figure 1).

Due to the high concentration of kraft lignin in the reaction, it was not completely solubilised at pH 10, so that it was found in two phases, an aqueous phase (A) and a solid phase (S). Samples of the non-treated (control) and laccase-treated lignins from the A and S phases were taken at 2 and 24 h to evaluate the changes in the Mw profile and in the phenolic and carbonyl contents of lignin over time (Figure 3). A progressive reduction in the amount of aqueous-soluble lignin (A phase) was observed during the reaction with laccase; the maximum absorbance of the lignin after 24 h of enzymatic reaction was half that of the control lignin (Figure 3A,B). Quite the opposite, the maximum absorbance of the deposited lignin (S phase) was remarkably superior after laccase treatment than in the control (Figure 3C).

The Mw and polydispersity of lignin are two fundamental properties that strongly influence its valorisation [29]. Kraft lignins extracted from *Eucalyptus globulus* exhibit Mw between 1300 and 32,000 Da [30]. The Mw profile of the eucalyptus lignin in the A phase was slightly broadened (towards higher and lower lignin moieties) after 2 h of reaction with laccase (Figure 3A), suggesting the simultaneous occurrence of depolymerisation and polymerisation reactions. Nonetheless, the most remarkable change was observed in the profile of the laccase-treated lignin in the S phase, with an important peak of lower Mw moieties not observed in the corresponding control that evidenced the breakdown of the lignin polymer (Figure 3C).

The phenolic content of the control lignin in the A phase rose with time (Figure 3D), in correlation with the progressively improved water-solubilisation of kraft lignin at pH 10. Above this, the enzyme significantly increased the phenolic content of lignin in the A phase (Figure 3D), in concordance with the depolymerisation/demethylation observed in the first experiment. In contrast, the phenolic content of the control lignin in the S phase was lower and it was halved after 24 h of reaction with laccase (Figure 3D). The latter suggests polymerisation and correlates with the observed increment of lignin deposited in the S phase after laccase treatment. These results showed that the dose of enzyme in the reaction influences the balance between cleavage and condensation reactions over time, with repolymerisation being favoured when high doses of enzyme are used (as in this second assay). This was confirmed in an additional experiment where the changes in the lignin phenolic content produced by different doses of laccase C-LeB were monitored at shorter reaction times (see Appendix A).

As for the content of carbonyl groups, it increased in all lignin samples after oxidation with laccase (Figure 3E), in agreement with the results from the first experiment. The carbonylic content of the lignin solubilised at pH 10 (A phase) augmented throughout the enzymatic reaction. This effect was also observed in the lignin in the S phase after 24 h of enzymatic treatment, even though in this case the content of carbonyl groups of the control lignin was already very high. As mentioned above, this increase in carbonyl groups is a well-known event caused by the oxidation of the lignin units’ side-chains by laccase. The oxidised lignin units can later undergo Cα–Cβ and aryl–alkyl cleavage [19], producing the release of oxidised monomeric compounds (aldehydes, ketones, acids), together with the formation of quinoid structures responsible for the observed darkening of the laccase-treated lignin [31].

#### 3.2.1. Lignin Mass Balance

After 24 h of reaction, the laccase-treated (L) and control (C) lignins soluble at pH 10 (and therefore found in the A phase) or deposited in the S phase were dried and resuspended in HCl, resulting in the corresponding acid-soluble (as) and acid-insoluble (ai) fractions (Figure 4). 

Of the total of 2.5 g of eucalyptus kraft lignin used in the assay, roughly half of the control lignin was found in the aqueous phase (C-A) and the other half precipitated in the S phase (C-S). The lignin in solid phase was significantly increased after the laccase treatment (L-S, 64% of total lignin), almost all of which was acid insoluble (L-Sai, 63.5%) (Figure 4). As for the lignin in aqueous phase, around half of the control lignin was insoluble in acid (C-Aai) and the other half was acid-soluble (C-Aas), whereas the majority of the laccase-treated lignin in the A phase was acid-insoluble (L-Aai, 28% of total lignin). The total mass balance for the control and laccase-treated lignins split in the different fractions (mg) were close to the theoretical value (2.5 g); see Appendix A.

#### 3.2.2. Chemical Characterisation of the Lignin Fractions Obtained after Acid Resuspension and Precipitation

As the majority of lignin was in the acid-insoluble fractions, the Mw and chemical composition of these fractions were analysed using GPC and NMR, respectively. In addition, the composition in simple monomers of the acid-soluble fractions was determined using GC/MS.

##### Determination of Molecular Weight

The acid-insoluble lignin fraction from the S phase after laccase treatment (L-Sai) was more polymeric than the corresponding control (C-Sai), as revealed by the higher Mw and Mn (Figure 5). The acid-insoluble lignin fraction from the A phase after laccase treatment (L-Aai) also showed an enrichment in high-Mw lignin moieties compared to the control (C-Aai). However, the PDI (Mw/Mn) was notably increased due to the significant increase in lower-Mw moieties, which evidenced depolymerisation of lignin by the enzyme. It seems that repolymerisation of the oxidised lignin products would reduce the solubility of the lignin moieties of higher Mw at pH 10, thus enriching the content of polymeric lignin in the S phase after 24 h of laccase treatment, whereas the majority of low-Mw oligomers or monomers would remain soluble in the reaction mixture.

##### Changes in Lignin Substructures as Observed in 2D-NMR Analyses

The chemical composition and structure of the four acid-insoluble lignin fractions were analysed with 2D-NMR (Figure 6). It was observed that lignin in the A phase after treatment with laccase (L-Aai) presented a markedly higher Syringyl/Guaiacyl ratio, while the content of β–β′ (epiresinol) decreased compared to the control (C-Aai). This may be related to the condensation of G units, which is in accordance with the increase in Mw together with the decrease in β–β′ substructures. The acid-insoluble fraction of S lignin treated with laccase (L-Sai) presented a notably higher content of β–O–4 linkages and higher content of β–5 linkages than the control (C-Sai). This increase in β–O–4 and β–5 linkages evidenced an enrichment of polymeric lignin [32] in the S phase, probably due to repolymerisation of the lignin products from laccase oxidation. Conversely, the β–β’ content of lignin notably decreased after the laccase treatment, in accordance with the aforesaid polymerisation reactions.

##### NMR Determination of Phenolic Hydroxyl and Carbonyl Groups

The analysis using ^31^P-NMR of the acid-insoluble fractions showed a decrease in free OH phenolic groups after laccase treatment compared to the control (Table 1). This decrement was more prominent for the lignin in the A phase (L-Aai) than in the S phase (L-Sai). These results contrast with the increase in phenolic OH content shown in Figure 3D. One possible explanation could be that the majority of molecules with free phenolic OH groups would migrate to the acid-soluble fraction (see next subsection).

#### 3.2.3. Analysis of Simple Phenols in the Acid-Soluble Fractions

The composition of simple lignin-derived phenols (identified with GC/MS after silylation) of the acid-soluble fractions of the control and laccase-treated lignins from phases A and S are shown in Appendix A. Laccase modified the relative abundance of the different phenolic compounds. In general, the content of the phenolic monomers was lower than in the corresponding controls, except for syringaldehyde. Particularly remarkable is the abundance of the phenols in the C-Aas sample, which suggests a significant content of low-Mw phenolic compounds in the kraft lignin soluble at pH 10 and later solubilised in acid. During the reaction with laccase, these phenols are oxidised and most of their radicals are involved in coupling reactions and, therefore, incorporated in more polymeric lignin, which is deposited in the solid phase.

To sum up, in this second assay, we obtained pieces of evidence for both lignin depolymerisation and polymerisation by laccase. The enzyme raised the phenolic and carbonyl content of the lignin soluble at pH 10, in correlation with the depolymerisation/demethylation observed in the first experiment (Section 3.1), whereas the remarkable increase in the lignin deposited in the S phase and the strong decrease in its phenolic content suggests polymerisation. The latter was supported by the more polymeric nature of the acid-insoluble lignin from the S phase (the major lignin fraction) after laccase treatment (higher Mw and Mn and higher content of β–O–4 linkages and β–5 linkages). Apparently, the coupling of lignin-oxidised products would reduce the solubility of lignin moieties with higher Mw at pH 10, thus enriching the content of polymeric lignin in the S phase after laccase treatment, whereas the majority of small oligomers and monomers would remain soluble.

### 3.3. Bench-Scale Trials in Bioreactor Coupled to a Membrane Separation System

Finally, with the aim to facilitate the depolymerisation of lignin by the alkaliphilic fungal laccase, the enzymatic reaction was carried out in a reactor coupled to a membrane separation system. The bench-scale run was performed on the same eucalyptus kraft lignin (25 g/L) with 5 U/g of Li10 alkaliphilic fungal laccase at pH 10 and 30 °C for 2 h. The resulting lignin fractions, a retentate and a permeate, were subsequently characterised with GPC, ^31^P-NMR, and 2D-NMR. The changes produced by the enzyme (L) were compared with the results obtained from a run performed under the same conditions without the enzyme (control, C). The average Mw of the lignin fractions (permeate and retentate) resulting from the bench-scale run with the fungal laccase were smaller than those of their corresponding controls. In particular, the Mw of the retentate was reduced by 4% and the permeate by 15% after reaction with the laccase (Table 2).

In general, the ^31^P-NMR analysis showed that the content of phenolic, carboxylic, and aliphatic hydroxyl groups of the retentate after the enzymatic reaction was very similar to that of the control, but significantly lower for the permeate (Table 2). This significant decrement in phenolic hydroxyls could be related to the complete oxidation of low-Mw lignin degradation products and simple compounds to form the corresponding quinones. In addition, phenol radicals formed during the enzymatic oxidation and degradation of lignin could be still stable long enough to be able to recouple in permeate through hydroxyl moieties after their removal from the reactor.

The 2D-HSQC NMR analysis showed that the total content of lignin linkages (per 100 aromatic units) in the permeate and the retentate obtained after reaction with the fungal laccase was lower than in their corresponding controls (Figure 7). This decrement was more pronounced in the permeate (14%), in correlation with the aforementioned 15% reduction in permeate Mw observed using GPC after the enzymatic treatment (Table 2). The capability of the high-redox potential alkaliphilic laccase, Li10 [15], to depolymerise kraft lignin in the absence of redox mediator compounds was therefore directly demonstrated thanks to the use of a membrane system that allows a reduction in overlapping repolymerisation reactions.

Other reported enzymatic routes developed to face the challenge of unlocking lignin depolymerisation are also based on the removal of the oxidised lignin products to prevent condensation reactions. They use either laccase-mediator systems combined with organic co-solvents to facilitate lignin solubilisation [33] or with the Lig system of *Sphingobium* sp. (etherases Lig E and Lig F, glutathione lyase Lig G) in enzymatic cascades [34], the fractionation and depolymerisation of lignin by laccase in deep eutectic solvents [35], or a membrane filtration system coupled to the enzymatic treatment (e.g., with Novozym 51003 laccase) [36]. However, in the latter case the reaction was carried out at an acidic pH and at a very low lignin concentration (0.6 g/L). In contrast, based on the METNIN™ (Metgen Oy, Kaarina, Finland) lignin refining technology principle [22], we took advantage of the high solubility of kraft lignin at alkaline pH, making mandatory the use of alkaliphilic laccases. We have been able to prove the enzymatic depolymerisation of lignin at 25 g/L, a lignin concentration similar to industrial conditions, without the need of adding redox mediator compounds or harsh organic solvents. Furthermore, it is envisaged that these results will be enhanced in the pilot METNIN™ lignin refining system (fully equipped with a set of membranes of different cut-offs), allowing us to obtain fractions with successively lower Mw (down to a pool of tri-, di-, and monomeric units) and reduced PDI [22].

The use of high-redox potential laccases may represent a significant advance over the use of low-redox potential bacterial laccases in the valorisation of technical lignins, particularly when valorisation is aimed at depolymerisation, given the superior ability of the former laccases to break down linkages with higher bond dissociation energies. This is especially important in the case of kraft lignins, which are highly condensed and therefore difficult to break down [37]. To explore this assumption, we performed another reaction with an alkaliphilic bacterial laccase in the same bench-scale setup and compared the results with those obtained with the fungal laccase. In fact, we carried out two runs with the bacterial laccase, respectively with five and twenty times higher activity doses than in the assay with the fungal laccase (due to the lower catalytic efficiency of the bacterial laccase). The corresponding retentate and permeate lignin fractions from both runs were analysed (Table 2). A strong increase in lignin Mw was observed in the retentate fractions obtained from the two laccase runs (L-1 and L-2). The Mw of the permeates after treatment with the bacterial laccase were also higher than that of the control. These results contrast with the Mw decrease of the lignin fractions observed after reaction with the fungal laccase. The remarkably higher Mw of the control retentate for the bacterial laccase compared with the control retentate for the fungal laccase can be explained by the eight times longer time of reaction with the bacterial laccase. Nevertheless, the enzyme doubled or tripled the lignin Mw of the control, evidencing a strong repolymerisation of lignin during the reaction with the bacterial laccase, even though depolymerisation should be favoured by the pH of 10.6 of the assay [22]. As regards the phenolic content, it was reduced as occurred with Li10 laccase, although now the effect was more pronounced, particularly in the permeate of the second laccase run. This reduction might be correlated with a more oxidised lignin (e.g., to form quinones), which is also supported by the higher amount of hydroxyl COOH groups after laccase treatment (Table 2).

Altogether, these results evidence the influence that the type of laccase and reaction conditions (pH, enzyme dosage, time of reaction, etc.) have on the balance between lignin depolymerisation vs. repolymerisation. They also highlight the higher efficiency of the high-redox potential fungal laccase over the bacterial laccase to depolymerise eucalyptus kraft lignin. It is envisaged that the dissimilar effect produced by the two laccase types would be even stronger for depolymerisation of softwood kraft lignins due to their more condensed nature and challenging depolymerization caused by their higher content in C–C linkages [38,39,40].

## 4. Conclusions

We address the challenge of contributing to the conversion of industrial lignins into valuable chemicals by demonstrating the feasible depolymerisation of kraft lignin using tailor-made alkaliphilic fungal laccases without the need for redox mediators. A profound chemical analysis of the treated lignin under different reaction conditions provided pieces of evidence on the enzymatic oxidation, demethylation, and depolymerisation of lignin. However, condensation of the oxidised products prevailed over depolymerisation in one-pot reactions, hampering the demonstration of the ligninolytic activity of the enzymes. For that, lignin treatment with the high-redox potential alkaliphilic laccase was finally accomplished in a reactor coupled to a membrane separation system. This setup allowed us to demonstrate the deconstruction of lignin by the high-redox potential enzyme, but not when a low-redox potential laccase was used. These results open a new scenario for the tailor-made fungal laccases as biocatalysts for kraft lignin valorisation.

## Figures and Tables

**Figure 1 polymers-15-04433-f001:**
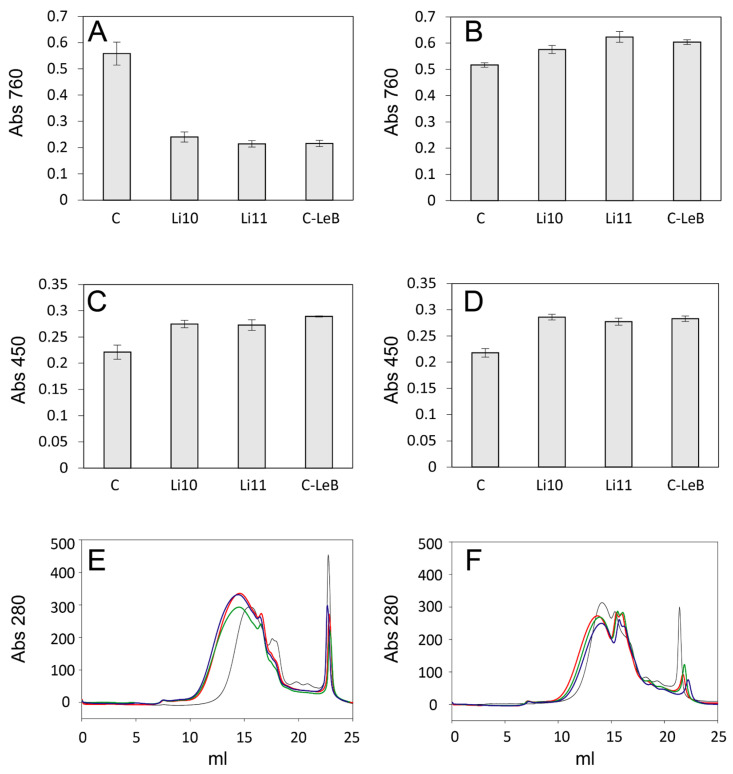
Changes in phenolic OH content measured with FCR (**A**,**B**), carbonylic content measured with Brady reagent (**C**,**D**), and Mw distribution with SEC (**E**,**F**) of eucalyptus kraft lignin after 24 h treatment with the three alkaliphilic laccase variants at pH 9 (**A**,**C**,**E**) and pH 10 (**B**,**D**,**F**). Control (black), Li10 (red), Li11 (green), and C-LeB laccases (blue). Each point represents the average of three replicates.

**Figure 2 polymers-15-04433-f002:**
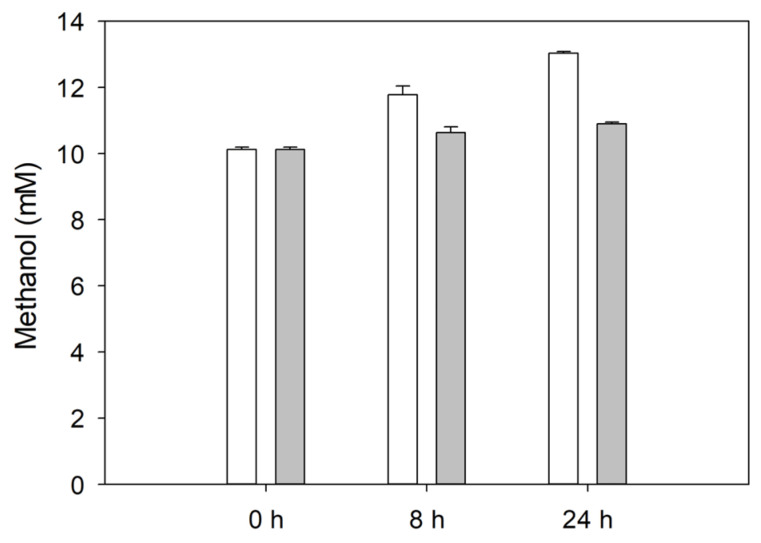
Methanol released from C-LeB laccase-treated (white) and non-treated (grey) kraft lignins at pH 10 measured using the Purpald assay. Error bars indicate the standard deviation of three replicates.

**Figure 3 polymers-15-04433-f003:**
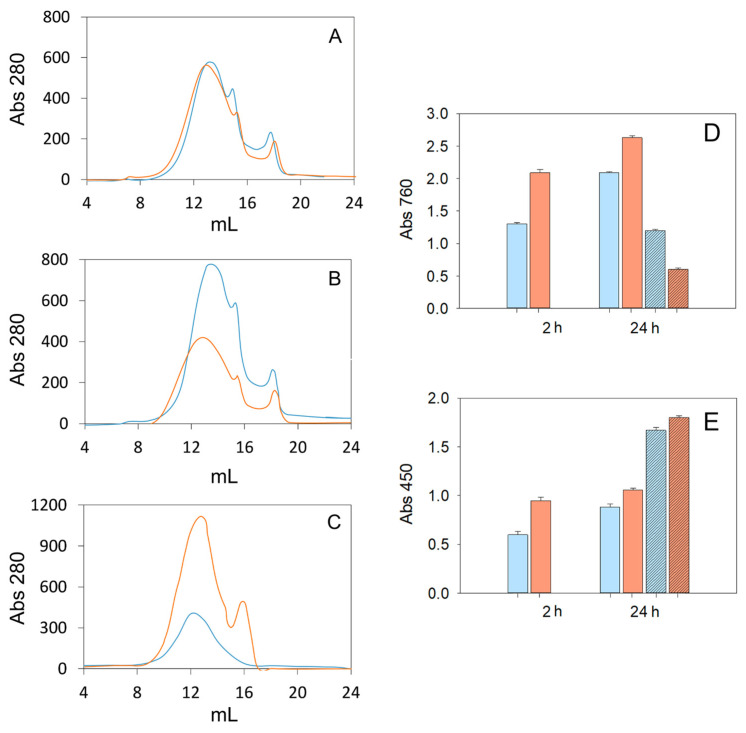
Changes in the Mw profile (**A**–**C**), and in the phenolic (**D**) and carbonyl (**E**) contents of control (blue) and laccase-treated (orange) lignins at pH 10. Shown are the Mw profiles of lignin in the aqueous phase after 2 h (**A**) and 24 h (**B**) of reaction, and after 24 h for the lignin in solid phase (**C**). Changes in phenolic and carbonyl contents are shown as plain bars for the lignin in the aqueous phase (**D**,**E**) and as striped bars for the lignin in the solid phase after 24 h (**D**,**E**). Error bars indicate the standard deviation of three replicates.

**Figure 4 polymers-15-04433-f004:**
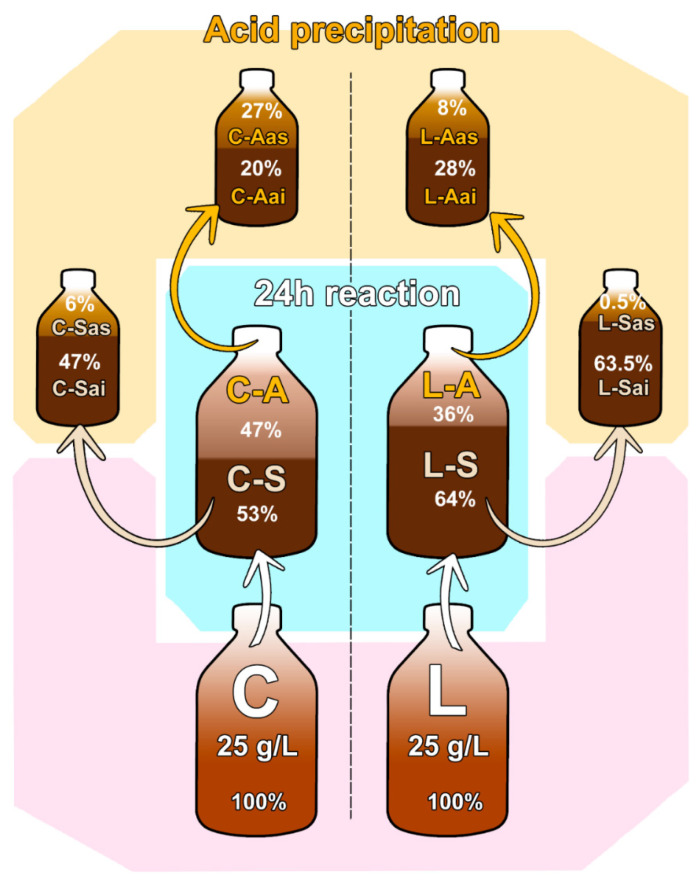
Eucalyptus kraft lignin fractions obtained after treatment (25 g/L) with laccase (L) or without enzyme (control, C) at pH 10 for 24 h. Lignins dissolved in the aqueous phase (A) or deposited in the solid phase (S) were dried and resuspended in acid, giving rise to the corresponding insoluble (ai) and acid-soluble (as) fractions.

**Figure 5 polymers-15-04433-f005:**
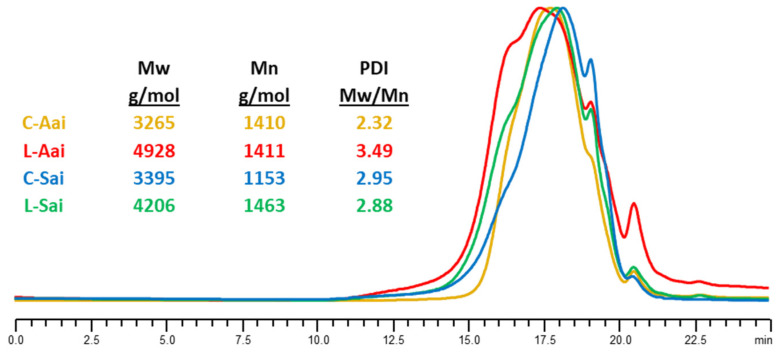
Changes in Mw of acid-insoluble lignin fractions from A and S phases of laccase-treated (L) and non-treated (C) samples. The colour of the chromatograms correspond to the colour codes assigned to the four lignin fractions.

**Figure 6 polymers-15-04433-f006:**
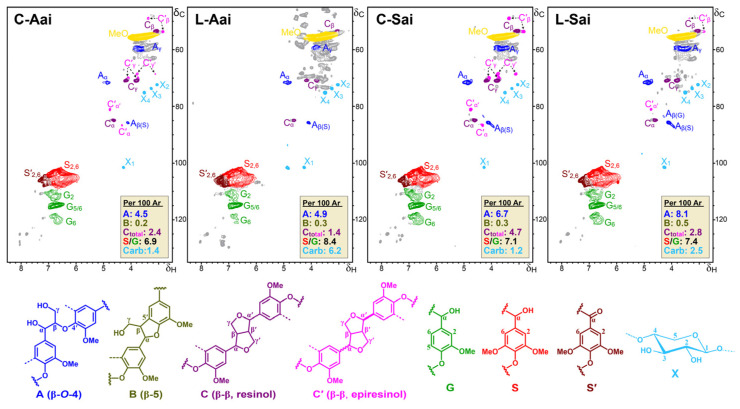
2D-HSQC NMR spectra of the acid-insoluble fractions from the aqueous and solid phases of control (C-Aai, C-Sai) and laccase (C-LeB)-treated (L-Aai, L-Sai) lignins. The identified structures are colour-coded to match their assignments in the spectrum.

**Figure 7 polymers-15-04433-f007:**
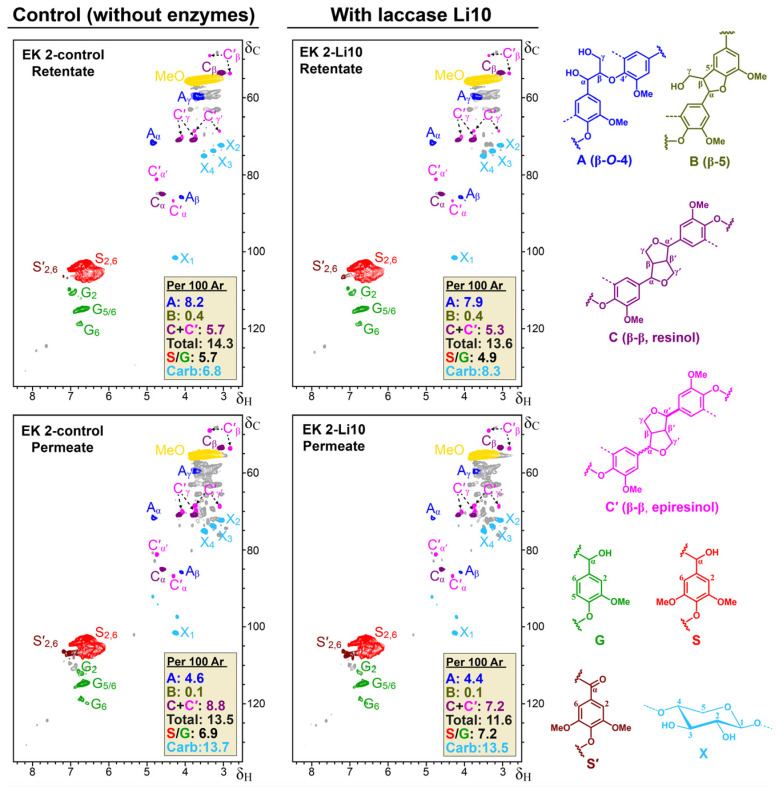
2D-HSQC NMR spectra of retentates and permeates obtained after treatment of eucalyptus kraft lignin in bioreactor with a membrane separation system without (control) and with Li10 laccase. The identified lignin substructures are shown on the right. Total values in boxes refer to the sum of all lignin linkages (A + B + C). Note: Signals of β-5 phenylcoumaran (B) are observed when intensities of the spectra are augmented.

**Table 1 polymers-15-04433-t001:** Contents of phenolic (Ph-OH) and carboxylic OH groups (mmoL/g) determined using ^31^P-NMR of the acid-insoluble lignins from A and S phases after 24 h treatment with C-LeB laccase (L) compared with the control (C). Cholesterol (Ch) and n-hydroxy-5-norbornene-2,3-dicarboximide (NH) were used as standards for quantification.

Sample	Ph-OH		Carboxylic-OH	
	Ch	NH	Ch	NH
C-Aai	4.12	3.60	0.65	0.55
L-Aai	2.81	2.50	0.50	0.38
C-Sai	4.28	3.60	0.25	0.20
L-Sai	3.98	3.19	0.27	0.21

**Table 2 polymers-15-04433-t002:** Average Mw (Da) and polydispersity (PDI) values determined using GPC, and content (mmoL/g) of phenolic (PhOH), carboxylic (COOH), and aliphatic (AliphOH) hydroxyl groups determined with ^31^P-NMR, of the permeate and retentate lignin fractions obtained from the treatment of eucalyptus kraft lignin with alkaliphilic laccases in a bench-scale membrane separation system. Enzyme dosage: 5 U of fungal laccase/g lignin; 25 U/g (run 1) or 100 U/g (run 2) of an alkaliphilic bacterial laccase.

Sample	Mw	PDI	PhOH ^a^	COOH ^a^	AliphOH ^a^	Total-OH ^a^
Lignin	2324	2.27	3.94 (3.77)	0.41 (0.41)	1.97 (1.93)	6.32 (6.10)
**Fungal laccase**						
Retentate C	5124	2.78	2.59 (2.98)	0.44 (0.54)	1.40 (1.60)	4.43 (5.12)
Retentate L	4940	3.07	2.55 (2.75)	0.43 (0.48)	1.43 (1.55)	4.40 (4.78)
Permeate C	1875	2.05	3.98 (2.57)	1.09 (0.81)	2.56 (1.78)	7.63 (5.16)
Permeate L	1595	1.92	2.58 (1.67)	0.82 (0.56)	1.92 (1.28)	5.32 (3.51)
**Bacterial laccase**						
**Run 1**						
Retentate C	10,482	4.0	2.95	0.63	1.82	5.4
Retentate L-1	20,383	7.6	2.62	0.73	1.68	5.03
Permeate C	3313	2.7	3.40	0.86	1.84	6.1
Permeate L-1	4935	3.5	2.47	1.00	1.61	5.08
**Run 2**						
Retentate C	10,482	4.0	2.95	0.63	1.82	5.4
Retentate L-2	37,498	13.6	1.95	0.67	1.41	4.03
Permeate C	3313	2.7	3.40	0.86	1.84	6.1
Permeate L-2	3892	3.1	1.71	0.97	1.32	4

^a^ Two internal standards, cholesterol and NHND (in parentheses), were used.

## Data Availability

The data presented in this study are available within the article or the Appendix A.

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
