# Peer review of "Depolymerisation of Kraft Lignin by Tailor-Made Alkaliphilic Fungal Laccases"

_polymers, 2023, doi:10.3390/polym15224433_

Round 1

Reviewer 1 Report

Comments and Suggestions for Authors

Interesting manuscript. Some of the analysis techniques are a bit unusual, but it appears that everything is described well. I do not see any problems with publishing this.

Author Response

We appreciate the reviewer’s positive comment.

Reviewer 2 Report

Comments and Suggestions for Authors

Review for polymers-2682592

The paper is devoted to the study of Kraft lignin degradation by laccases in alkaline media. This is the interesting problem, but the results obtained looks like insignificant.

Fig. 1, B and D, show very low activity of laccases used at pH 10 in comparison with the control experiment, less then 10-25%. 

Lines 259-261. The methanol released from the laccase-treated lignin increased with reaction time compared to the control, indicating that lignin was demethylated by the enzyme (see Supplementary data). This conclusion should be illustrated quantitatively in the manuscript, but not in supplementary. Moreover, it is not available for reviewer:  Supplementary data to this article contains two tables and three figures that can be found online. www.mdpi.com/xxx/.   

Table 2. Run 1 and run 2 in the table are not shown clearly.

For the fungi laccase Ret L / Per L > Ret C / Per C (3.1 and 2.7 correspondingly). This indicates that Laccase does not destruct lignin even using the METNINTM technology.

Moreover, all the fraction of lignin treated with bacterial laccase and even of control experiments have Mw exceeded the Mw of initial lignin. This ratio is not understandable and do not confirm decreasing the lignin molecular weight.  

The interval of molecular masses in Table 2 (1600-37500) contrasts with a little differences between GPC lines in Fig. 1-3. 

To conclude, the results of Table 2 do not confirm the conclusion that the combination of high-redox potential alkaliphilic laccase with a filtration system (METNIN™) allowed to demonstrate the deconstruction of lignin.

And a general imagine on the manuscript. The goals written in the paper are not attained, the registered effects are very small.

Author Response

REVIEWER 2

The paper is devoted to the study of Kraft lignin degradation by laccases in alkaline media. This is the interesting problem, but the results obtained looks like insignificant. 

Fig. 1, B and D, show very low activity of laccases used at pH 10 in comparison with the control experiment, less than 10-25%.  

Response: A 10-20% modification of the phenolic OH (Ph-OH) and carbonyl content of lignin by the three enzyme variants is not necessarily a negligible result. The results are statistically supported (they come from three replicates). Moreover, they are supported by the changes in lignin’s Mw distribution profile observed by size-exclusion chromatography. In addition, the main goal of this first experiment is achieved. On one hand, it shows that the three enzyme variants oxidize (Fig 1D) and modify (Fig 1F) the lignin at pH 10, all of them in a similar way. On the other hand, it shows the influence that pH has on the fate of the oxidized radicals in the reaction. At pH 9, the strong decrease of Ph-OH correlates with the strong repolymerization observed by SEC, whereas at pH 10, the observed tendency is a slight increase of the Ph-OH content together with the appearance of new lignin species with lower Mw.

Lines 259-261. The methanol released from the laccase-treated lignin increased with reaction time compared to the control, indicating that lignin was demethylated by the enzyme (see Supplementary data). This conclusion should be illustrated quantitatively in the manuscript, but not in supplementary. Moreover, it is not available for reviewer:  Supplementary data to this article contains two tables and three figures that can be found online. www.mdpi.com/xxx/.   

Response: We are sorry if the reviewer could have problems accessing to the supplementary material. However, we uploaded the corresponding pdf file as indicated. Anyway, following the reviewer’s suggestion we have moved the figure illustrating the enzymatic demethylation of lignin to the main manuscript (now Figure 2).  On the other hand, we understand that the journal will provide the details for the following link “www.mdpi.com/xxx/” to make the revised Supplementary information available to the reader.

Table 2. Run 1 and run 2 in the table are not shown clearly. 

Response: We have corrected the table as suggested to make it clearer (see revised Table 2).

For the fungi laccase Ret L / Per L > Ret C / Per C (3.1 and 2.7 correspondingly). This indicates that Laccase does not destruct lignin even using the METNINTM technology.

Response: From our point of view, the ratios indicated by the reviewer are not relevant since the values shown in Table 2 do not correspond to the amount of lignin present in the retentate or permeate fractions, but to the average mass/size (Mw) of the lignin fractions (from GPC analysis).

The important result shown here is that when comparing the average Mw of the retentate and permeate fractions obtained from the reaction with laccase or from the control, we observed a decrease in Mw in both fractions (4% in the retentate and 15% in the permeate) due to the action of the fungal laccase (first column Table 2). These values have been added in the revised text (Lines 459-460).

According to the rationale of the reviewer, Ret L / Per L > Ret C / Per C means there is no depolymerization. However, in case of complete depolymerization of lignin by the enzyme, the Mw of the permeate would be strongly decreased. Therefore, the Ret L/ Per L ratio could be very much increased (being much higher than in the control), but this wouldn’t mean absence of lignin deconstruction, just the opposite.

Moreover, all the fraction of lignin treated with bacterial laccase and even of control experiments have Mw exceeded the Mw of initial lignin. This ratio is not understandable and do not confirm decreasing the lignin molecular weight.   

Response: Table 2 shows, precisely, the different effects of enzymatic reactions carried out with a fungal laccase or a bacterial laccase as compared with the corresponding controls (carried out under the same conditions without enzyme). In the first case, we observed depolymerization, while in the second case (in the two runs with the bacterial laccase), we observed a strong repolymerization of lignin.  The remarkably higher Mw of the control of the bacterial laccase (as compared with the control for the fungal laccase) can be explained by the longer time of reaction with the bacterial laccase. Nevertheless, the enzyme doubled or tripled the lignin Mw of the control retentate, thus evidencing a strong repolymerization of lignin during the reaction with the bacterial laccase, even though depolymerization should be favored by the pH 10.6 of the assay. We have rewritten this part of the text explaining the experiment better -especially the part of the bacterial laccase- and the results obtained (lines 450-460, 510-537).

The interval of molecular masses in Table 2 (1600-37500) contrasts with a little difference between GPC lines in Fig. 1-3.

Response: We are sorry, but there has been a misunderstanding. Fig. 1-3 do not show the results from GPC analyses. Only Fig. 4 (currently Fig. 5) shows the chromatograms from GPC analysis, although it shows the results from an experiment not related to that illustrated in Table 2. The results shown in Fig. 4 (currently Fig. 5) come from the second experiment carried out with fungal laccase at a lab-scale (in a bottle) using 25 g/L of lignin. As lignin was not completely solved at pH 10, it was found in two phases (solid, S, and aqueous, A); the GPC results correspond to the Mw profiles of the acid-insoluble fractions of these lignins (after laccase treatment compared with their controls). In contrast, Table 2 illustrates the results from bench-scale assays carried out either with the fungal or the bacterial laccase in quite different conditions, since they were performed in a bioreactor coupled to a membrane separation system (under specific conditions for the two enzymes).  Thus, the initial amount, chemical characteristics, and Mw of lignin cannot be comparable between these two assays.

Moreover, the mentioned Mw value of 37500 was found only when the bacterial laccase was utilized in the bench-scale trial (Table 2), and not with the fungal one. As explained above, and also explained in the revised text, the reaction conditions have a remarkable influence in the balance between lignin depolymerization vs repolymerization, as well as the type of laccase used, highlighting the higher efficiency of the high-redox potential fungal laccase over the bacterial laccase to depolymerize Eucalyptus kraft lignin (lines 521-537).

To conclude, the results of Table 2 do not confirm the conclusion that the combination of high-redox potential alkaliphilic laccase with a filtration system (METNIN™) allowed to demonstrate the deconstruction of lignin. 

Response: We are sorry to disagree. In section 3.3 of this article, it is clearly demonstrated how the average Mw of the permeate lignin after laccase treatment is 15% lower than that of the control permeate. The molecular size of the retentate was also 4% lower than that of the control (Table 2). In addition, these GPC results are in accordance with those found in the 2D-NMR HSQC analyses, that show how after reaction with the fungal laccase, the total content of lignin linkages (per 100 aromatic units) decreases in the permeate, and even in the retentate (5%),  as compared to the corresponding controls. The decrement in total linkages was again more pronounced in the permeate (14%), in correlation with the aforementioned 15% reduction in permeate Mw observed by GPC.

Still, we have edited the wording of the text to enhance the clarity of these primary conclusions. See Lines 459-460, 477-480.

And a general imagine on the manuscript. The goals written in the paper are not attained, the registered effects are very small.

Response: We are sorry to disagree. In this study, we aimed to prove the ligninolytic activity of the tailor-made alkaliphilic fungal laccases on kraft lignins, without addition of harsh organic solvents and/or synthetic redox mediators This have been proved, and we expect an enhancement of the benefits obtained in larger scale reactor trials that will be carried out in a near future.

Reviewer 3 Report

Comments and Suggestions for Authors

The article describes use of laccase for depolymerization of kraft lignin and its chemical modification. The analysis is extensive, and the idea is original. However, organization of the article needs to be considered to follow the flow of information easily. The chemical modification section should be better placed along with its prospects in synthesis of new or different compounds.

There are many language errors including number disagreement, verb confusion and redundant words for instance in the lines 10-11, 14.

In abstract, the rationale of this study should be given clearly. If authors previously developed fungal laccase preparation for depolymerization of kraft lignin, then the significance of this study needs to be elaborated.

In introduction, highlight the significance of alkali-active laccase and its broad applications.

The sources of laccase stated in section 2.2 and 2.3 are apparently different. Without any detail in introduction or in method, it is difficult to conceive this difference.

Some of the references are incomplete.

Comments on the Quality of English Language

There are many language errors including verb confusion that need to be fixed.

Author Response

REVIEWER 3

The article describes use of laccase for depolymerization of kraft lignin and its chemical modification. The analysis is extensive, and the idea is original. However, organization of the article needs to be considered to follow the flow of information easily. The chemical modification section should be better placed along with its prospects in synthesis of new or different compounds.

Response: We appreciate the reviewer’s comment, and are aware that the work compiles quite a few assays and chemical analyses that might be difficult to follow. However, we organized the paper according to the workflow and the rationale behind the assays, each one having their respective chemical analyses to evaluate the changes produced in the lignin polymer by the enzyme(s). In this study, we aimed to prove the ligninolytic activity of the tailor-made alkaliphilic fungal laccases on kraft lignins, without addition of harsh organic solvents and/or synthetic redox mediators. There was no particular aim towards obtaining specific aromatic compounds, but rather to show the potential of these recently developed enzymes as biocatalysts for the valorisation of kraft lignins.

In the first assay (subsection 3.1), we addressed the effect of pH on lignin transformation by three alkaliphilic fungal laccase variants, in small lab-scale, in one-pot reactions. This investigation provided compelling evidence for the similar oxidative effects of the three enzymes, and the depolymerisation tendency of lignin favoured at pH 10. Taking this into account, we carried out a second lignin treatment on a larger lab-scale with one of the fungal laccases (subsection 3.2). This involved higher enzyme dosage and increased lignin concentration (same as that used in METNIMTM process). Following an extensive chemical characterization of the obtained lignin fractions (detailed in subsection 3.2.2), we once again observed evidence of enzymatic oxidation and lignin depolymerization. However, our analysis revealed a prevalent repolymerization of the oxidized lignin products during one-pot reactions, hindering a conclusive demonstration of enzymatic lignin depolymerization. To overcome this limitation, we conducted the final assay (subsection 3.3) within a bioreactor coupled to a membrane separation system based on the METNINTM lignin refining principle. In this concluding experiment, we successfully demonstrated the deconstruction of the lignin polymer by the fungal laccase. In contrast, the bench-scale trials involving an alkaliphilic bacterial laccase primarily resulted in lignin repolymerization.

After considering a possible reorganization of the paper, we have decided to maintain the existing sections. However, we have edited the headings of Material & Methods, revised and re-numbered the headings (and subheadings) of Results and Discussion subsections 3.1-3.3, and revised their content as well as the conclusions, in order to clarify the specific objectives of each assay and to highlight the principal results and conclusions derived from the chemical analyses. See revised headings (lines 122, 227, 247, 291, 377-378, 383, 397, 414, 426 and 449) and text (lines 295-296, 437-448, 450-460, 477-480, 493-505, 533-537, 539-550).

There are many language errors including number disagreement, verb confusion and redundant words for instance in the lines 10-11, 14.

Response: We have corrected the above errors and carefully revised the language throughout the article.

In abstract, the rationale of this study should be given clearly. If authors previously developed fungal laccase preparation for depolymerization of kraft lignin, then the significance of this study needs to be elaborated.

Response: The abstract, and other sections of the manuscript, has been edited according to this comment and the previous comment.

In introduction, highlight the significance of alkali-active laccase and its broad applications.

Response: As suggested, we have also corrected the introduction by adding useful information (and references) about the biotechnological potential of the new alkaliphilic laccases. Lines 66-79.

The sources of laccase stated in section 2.2 and 2.3 are apparently different. Without any detail in introduction or in method, it is difficult to conceive this difference.

Response: We have now explained the differences in the revised text. Lines 295-296.

Some of the references are incomplete.

Response: We have carefully reviewed and updated the references (5, 16-20, 35).

Round 2

Reviewer 2 Report

Comments and Suggestions for Authors

Dear Authors, 

Thank you for the comprehensive responses. 

Best regards.

Reviewer. 
